# Probabilistic mathematical modelling to predict the red cell phenotyped donor panel size

**Denisse Best[1]\*, Kevin Burrage[2], Pamela Burrage[2], Diane Donovan[3], Shamila Ginige[1], Tanya Powley[1], Bevan Thompson[3], James Daly[1]**

**1** Australian Red Cross Lifeblood, Brisbane, Queensland, Australia, **2** School of Mathematical Sciences, Queensland University of Technology, Brisbane, Queensland, Australia, **3** School of Mathematics and Physics, University of Queensland, Brisbane, Queensland, Australia

\* dbest@redcrossblood.org.au

**Data Availability Statement:** All relevant data are within the paper and its Supporting Information files.

## Abstract

In the last decade, Australia has experienced an overall decline in red cell demand, but there has been an increased need for phenotyped matched red cells. Lifeblood and mathematicians from Queensland universities have developed a probabilistic model to determine the percentage of the donor panel that would need extended antigen typing to meet this increasing demand, and an estimated timeline to achieve the optimum required phenotyped (genotyped) panel. Mathematical modelling, based on Multinomial distributions, was used to provide guidance on the percentage of typed donor panel needed, based on recent historical blood request data and the current donor panel size. Only antigen combinations determined to be uncommon, but not rare, were considered. Simulations were run to attain at least 95% success percentage. Modelling predicted a target of 38% of the donor panel, or 205,000 donors, would need to be genotyped to meet the current demand. If 5% of weekly returning donors were genotyped, this target would be reached within 12 years. For phenotyping, 35% or 188,000 donors would need to be phenotyped to meet Lifeblood's demand. With the current level of testing, this would take eight years but could be performed within three years if testing was increased to 9% of weekly returning donors. An additional 26,140 returning donors need to be phenotyped annually to maintain this panel. This mathematical model will inform business decisions and assist Lifeblood in determining the level of investment required to meet the desired timeline to achieve the optimum donor panel size.

## Introduction

The International Society for Blood Transfusion Committee on Terminology for Red Cell Surface Antigens has recognized over 300 blood group antigens. The combinations of these blood group antigens vary between individuals so transfused red blood cells (RBC) have the potential to cause an immune response in recipients that lack certain antigens. Patients who have red cell antibodies that are considered clinically significant will require red cells that are negative for the corresponding antigen. Depending on the specific alloantibody in the recipient and the

**Funding:** The author(s) received no specific funding for this work.

**Competing interests:** The authors have declared that no competing interests exist.

prevalence of the blood group antigen in a donor population, delivery of compatible RBCs can be difficult [1]. Patients that have specific conditions or are on treatments that make red cell antibodies difficult to exclude, or where it is considered beneficial to reduce the risk of alloimmunisation may also require transfusion with red cells lacking one or more blood group antigens. The limited shelf life of red cell products and the reliance of blood donor appointments makes the estimation of the phenotyped inventory complex and challenging.

While there has been an overall reduction in the demand for RBC in Australia (approximately 20% since 2011/12), there has been a rise in the demand for phenotype matched red cells (approximately 50% over three years). As the sole provider of fresh blood components to the Australian population, the Australian Red Cross Lifeblood has a keen interest in identifying blood donors with a range of red cell phenotypes.

Lifeblood routinely performs ABO, Rh (D, C, E, c and e) and K typing to match blood donors and recipients. Extended antigen typing with antibody-based serological methods can be laborious, expensive, and limited to reagent antibody availability and is only performed on a proportion of repeat donors. When reagent antibodies are not available for phenotyping, DNA microarrays that target specific single nucleotide polymorphisms (SNPs) can also be used for extended blood group typing [2]. Blood donors who have been genotyped on previous donations will have historical genotype information that can predict the phenotype and be used without requiring re-testing [3].

Mass screening of blood donors with extended phenotyping and genotyping provides the opportunity to create and maintain a large database to meet demand for specific antigen-negative requirements. The additional cost of extended blood group phenotyping and/or genotyping generally precludes testing the entire blood donor panel. However, there is limited literature on the optimum percentage of the donor pool that is required to be tested to reliably meet the demand for phenotype matched red cells.

The majority of molecular blood group screening programs should be able to cover around 97% of all clinically important blood groups [4]. Portegysa et al. [5] describes one such regional donor registry established in Germany that provides matched blood products on demand.

The Canadian Blood Service (CBS) is responsible for providing blood to all areas of Canada, outside of Quebec, and has approximately 450,000 donors providing around 900,000 units of RBCs annually. It was estimated that 20–30% of these donors have some level of extended phenotyping, beginning with extended Rh (C, c, E and e) and K phenotyping and then on to Duffy ($Fy^a$ and $Fy^b$), Kidd ($Jk^a$ and $Jk^b$), S and s typing on subsequent donations if indicated [6]. CBS found that with a screening rate of 30%, it would take three to five years to achieve a sufficient registry to service their population [6]. Following on from this, they estimated an ongoing screening rate of 10–20% for their new donors to maintain a stable inventory. This analysis was based on a rare donor turnover of less than 10% each year, and the addition of related donors for each new rare donor discovered from the screening [6].

Similarly, Hema-Quebec implemented a strategy to create a database from genotyping 21,000 frequent blood donors over two years [7]. Their analysis demonstrated that this database should provide enough variation in blood group antigens to cover 95% of the 16,500 annual hospital requests for phenotyped RBCs. They also estimated that they would need to genotype 4,000 new donors each year to maintain this database.

The Bloodcenter of Wisconsin in the United States of America (USA) performed mass-scale genotyping on their blood donors over four years and by the final year of the program, 29% of all blood units on any given day had a known genotype. This allowed 99.8% of patient encounters to be resolved by their own database and inventory. The Bloodcenter of Wisconsin determined that each year, 4,000 donors would need to be genotyped to maintain this database [3, 8].

In the Netherlands, the Sanquin Bank of Frozen Blood (SBFB) runs a national typing program of blood donors. All donors are typed for ABO, Rh and K typing, with K positive donors being typed for k. In addition, 30% of donors were further typed for Fy[a], Fy[b], Jk[a], Jk[b], M, N, S and s and a further few were typed for Kp[a], C[w], Co[b], Wr[a], Lu[a], Le[a], Le[b] and P1. With this database, Sanquin were able to find compatible units in 95% of requested cases [9].

In collaboration with mathematicians from Queensland universities, Lifeblood conducted this project to develop a mathematical model to determine:

- the percentage of the Australian whole blood donor panel requiring additional phenotyping or genotyping, to ensure that sufficient donors with the commonly ordered antigen combinations are available to meet demand;

- the timescale for achieving the minimum percentage of phenotyped or genotyped donors;

- how this timescale may be reduced by increasing the percentage of inventory tested;

- the percentage of inventory to be tested to maintain the optimum number of whole blood donors phenotyped once the target has been reached.

There is very little modelling work in the literature to address this or similar blood demand problems, but we highlight work done by Blake and Clarke [6] that was primarily designed to evaluate the impact of frozen inventory. In their paper a two-phase approach was developed to determine how rare a blood type needed to be before freezing and the associated screening rates. Discrete event simulations were run with their model evaluating a single antigen at a time. When exploring scenarios based around 29 different antigens, these were treated as mutually exclusive, so that the underlying statistical distributions are Binomial.

A natural modelling approach is based around the Multinomial distribution, where the number of categories is greater than 2. The probabilities are associated with a list of "antigen combinations" based on the ABO blood group system, as well as a combination of 40 other blood group systems and over 300 different blood group antigens.

## Methods

To estimate the level of phenotyping and genotyping to meet clinical demand with at least 95% success percentage, we started with a set of individuals (donors) to be allocated (typed) across $r$ disjoint independent categories (antigen combinations). Each category was assigned a target (number of donors) and a probability of assigning an individual to that category. Then we predicted the number of individuals (donors) required to reach the targets under a random allocation, refer to Fig 1.

A crucial step in this process was the determination of the probability distribution. Whilst each antigen has its own probability distribution, these are generally population specific and the occurrence of each antigen in a particular antigen combination is not always independent. For example, the probability of a phenotype E-, Fy[a]-, Jk[b]-, K- and Lu[a]- may not be the product of the probabilities of the individual components. To address this issue, we used historical frequency information from previous Lifeblood requests to build a set of probabilities for each of the categories (combination of antigens) requested. The probabilities of the antigen combinations of interest were calculated by taking the frequency of a requested product (antigen combination) as a percentage of the total number of requests, refer to Fig 2. It was assumed that the historical request data, used to set targets for the allocation of donors to $r$ categories, was sufficiently comprehensive to determine reasonable estimates for the probabilities of each of these categories.

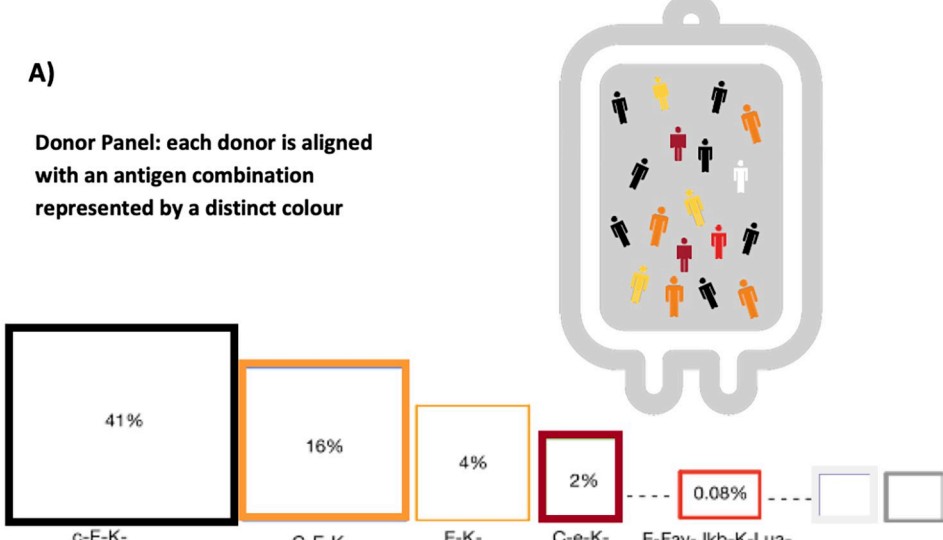

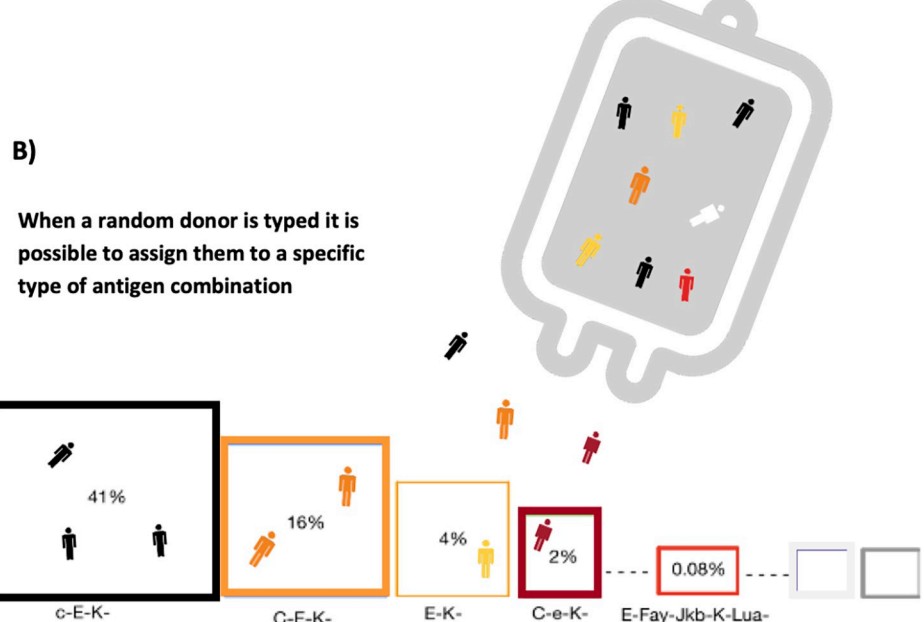

**Fig 1.** A) Donor panel and indicative probability distribution for r categories (shown as blood bags); B) Given the probabilities for each category, donors are assigned at random to the categories.

Each donor is assumed to be aligned with a unique category, that is has a combination of antigens unique to the donor. In the simulation *S* donors are allocated across *r* categories based on the category probabilities in Fig 2, the probability (frequency) of observing a donor of that category. Thus, to simulate *S* donors coming through the door at random the allocation to a category is achieved by a random process based on a Multinomial distribution (refer to Table 1). Many software packages implement this. Matlab can be used with command mnrnd

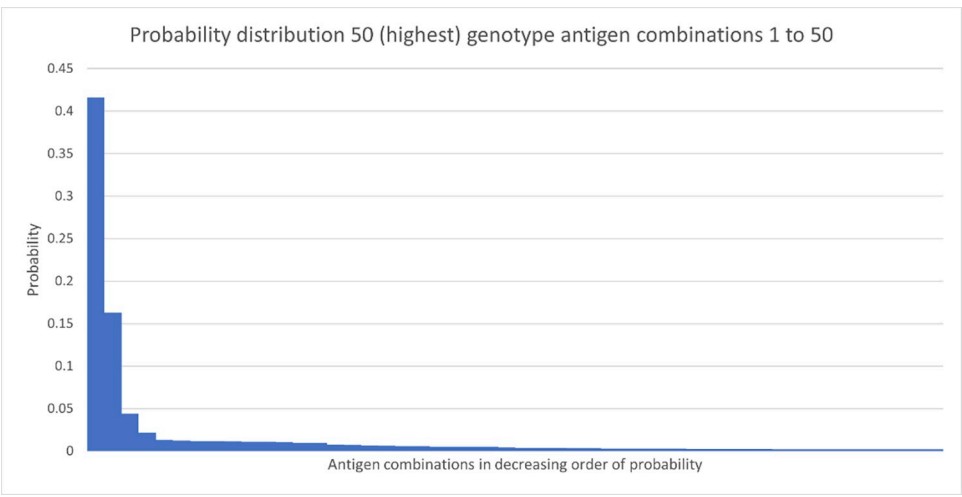

**Fig 2. Probabilities for 50 (highest) genotype antigen combinations.** The horizontal axis shows antigen combinations, and the vertical axis shows probabilities.

($S$,$p$), where $p$ is a vector of length $r$ the number of different categories. If after $S$ donors are allocated, the targets for all the categories are not reached, then the number of donors is increased until all the targets are met. To ensure the targets for the uncommon phenotypes are met, the simulation is continued until at least one donor is allocated to each category. However the Multinomial distribution ensures more donors are allocated to catagories with a higher probability. So, in the simulation, categories with greater probability will be allocated more donors, with the one donor case being the rarest situation for categories of low probability. This random process ensures that more donors are allocated to categories with higher probabilities, meeting these targets as well. In building the Multinomial distribution model we used the following framework:

Under a Multinomial distribution on the category types $A_1, \cdots, A_r$, with probabilities $p_1, \cdots, p_r \in [0,1]$ where $p_1 + \cdots + p_r = 1$, the probability of obtaining typed donors to match ($n_1, \cdots, n_r$) requests, where $\sum_{i=1}^{r} n_i = S$, is given by

$$P = Prob(n_1, \cdots, n_r) = \frac{S!}{\prod_{j=1}^{r} n_j!} \prod_{j=1}^{r} p_j^{n_j}.$$

Here $\prod_{j=1}^{r} p_j^{n_j}$ denotes the product of $r$ terms, $p_j^{n_j}$ over $j = 1, \cdots, r$, and $a! = a.(a-1).(a-2)\cdots 2.1$.

The following theorem underpins the mathematical model:

**Table 1. Notation to be used in the specification of the Multinomial distribution.**

| Glossary | |
|---|---|
| **Notation** | **Specification** |
| $S$ | Number of donors to be typed |
| $R$ | Number of distinct categories (unique identifies, products) |
| $A = (A_1, \cdots, A_r)$ | Vector containing the distinct categories |
| $Nvec = (n_1, \cdots, n_r)$ | Vector containing the number of typed donors in each category |
| $Pvec = (p_1, \cdots, p_r)$ | The vector of probabilities, one for each category |
| $Kvec = (k_1, \cdots, k_r)$ | Targets for the number of typed donors for each category |

**Theorem 1:** Given the above Multinomial distribution where $S$ donors are to be allocated to $r$ categories, $A_1, \cdots, A_r$, with probabilities $p_1, \cdots, p_r$, then the expected number of donors to be allocated to category $A_i$ is given by $\mathbb{E}[n_i]$, with variance $\mathbb{V}ar[n_i]$, where

$$\mathbb{E}[n_i] = p_i S,$$

$$\mathbb{V}ar[n_i] = p_i(1 - p_i)S, \quad i = 1, \cdots, r,$$

noting that the marginal distributions are Binomial distributions.

The vector of given targets (*Kvec*) and the sum of these targets (*K*), are used to estimate the size of $S$ so that all the components of the vector of targets are met with a certain probability. The size $S$ must be greater than or equal to $K$. However, it is very unlikely to be equal to $K$ as some of the Multinomial samples needed to match a given component of the target vector may be considerably greater than the individual targets. We estimated $S$ and the associated probability.

$$P_S = Prob(n_1 \geq k_1, \cdots, n_r \geq k_r)$$

where $\sum_{j=1}^{r} n_j = S$.

Intuitively if $S-K$ is positive but close to 0, we would expect the probability $P_S$ to be very close to 0. However as $S-K$ becomes larger, the probability of meeting the targets approaches 1.

One way of addressing this problem is to perform a number of Multinomial trials to estimate $S$ for a given set of targets and probabilities. Unfortunately, this is not a computationally feasible approach even for moderate sizes of $r$. However, when $S$ is large, the following theorem by Mallows (see also Esary, Proschan and Walkup)can be applied to find good approximations to these probabilities and good estimates of $S$.

**Theorem 2:** The following lower and upper bounds on $P_S$ hold

$$1 - \sum_{j=1}^{r} Prob(n_j < k_j) \leq P_S \leq \prod_{j=1}^{r} Prob(n_j \geq k_j) \leq exp(-\sum Prob(n_j < k_j)).$$

**Proof:** Mallows [10, 11]

Here the individual probabilities $Prob(n_j < k_j)$ and $Prob(n_j \geq k_j)$ are the marginal (binomial) probabilities and are very quick to compute. Importantly, as $S$ becomes large and the probability $P_S$ gets closer to 1 then these bounds get close together, and so provide an excellent approximation to the probability $P_S$. We chose a high probability $P_S$ of 0.95.

The sample size was approximated with the lower bound being greater than 0.95

$$P_S \geq 1 - \sum_{j=1}^{r} Prob(n_j < k_j) > 0.95. \tag{1}$$

Once we have estimated $S$ in this manner, we validate the results by running a set of Monte Carlo Multinomial samples to determine if indicative request targets can be met.

For a given $S$ we simulate from the appropriate Multinomial distribution and record the vector *Nvec* = $(n_1, \cdots, n_r)$ as the number of donors allocated to each category. The number of negative terms in the difference *Nvec*−*Kvec* = $(n_1-k_1, \cdots, n_r-k_r)$ represents the number of times a target is not attained.

It is well known that one area in which standard Monte Carlo simulations do not work particularly well are in rare event simulation [12]. In this case many simulations may be needed to estimate these rare events and this can become computationally extremely burdensome. New approaches have been developed such as the cross entropy method [12] that can be represented

as a stochastic search algorithm that can address the issue of rare event estimation, but implementations are not trivial. However, in our case we have access to tight upper and lower bounds on the multinomial probability of meeting targets with high probability through the theory of Mallows. Thus rare event estimation of rare antigen combinations is just a matter of computing marginal (binomial) probabilities and avoids the issues associated with rare event simulation through Monte Carlo approaches. However, to cross validate results we generated 10,000 Monte Carlo Multinomial samples and computed the number of times that *Nvec−Kvec* has a negative component in any position. The number of failures to meet targets was calculated as a percentage of the number of categories *r*. Taking one hundred minus this value gave the minimum percentage of success in attaining the targets. The Monte Carlo Multinomial sampling was repeated using increasing values of *S* until at least 95% success percentage was obtained in all 10,000 Monte Carlo samples. This was compared with the estimates computed by the approach of Mallows, described in Theorem 2.

## Determination of testing regimes needed to meet targets

Using the described mathematical framework, formulae were developed to estimate the testing schedule (timeline) needed to meet the required targets. These estimates were based on bounds given in Theorem 2.

We use the average number of screened donors available in a year to estimate the percentage of requests that can be met. Using the notation in Table 2, the average number of screened donors in year 1 is

$$D_1 = (1 - m)D_0 + q(1 - m/2).$$

Here the choice of m/2 is based on the mean of a uniform distribution of donors throughout the year. That is, there are *q* donors screened over the course of the year and thus, on average, a newly screened donor has been registered for six months at the end of the first year. Accordingly, the non-return rate has been adjusted by *m/2* to reflect this fact. Setting $M = q(1 - m/2)$, the average number of screened donors in year 2 is

$$D_2 = (1 - m)((1 - m)D_0 + M) + M = (1 - m)^2 D_0 + M(1 - m) + M.$$

By induction, the average number of screened donors in year *y* is

$$D_y = (1 - m)^y D_0 + M\big((1 - m)^{y-1} + (1 - m)^{y-2} + \cdots + (1 - m)^1 + 1\big)$$
$$= (1 - m)^y D_0 + \frac{M}{m}(1 - (1 - m)^y).$$

This formula enabled calculation of either *q* or the number of years *y*, given knowledge of the targeted number of donors $D_y$ and the initial number of donors already screened $D_0$.

**Table 2. Notation for calculation of timelines.**

| Glossary | |
|---|---|
| **Notation** | **Specification** |
| $D_0$ | Number of donors already screened |
| $D_y$ | Average number of screened donors in the system in year *y* |
| *Q* | The number donors screened per annum |
| *m* | The fraction of non-returning screened donors per annum as a decimal |

## Results

The mathematical Multinomial model was used to analyse supply and demand data for phenotyped red cells. Lifeblood's phenotyped red cell demand for approximately 27000 requests was analysed. The requests for antigen negative combinations achieved through extended phenotyping by serology or genotyping to predict the phenotype for the following antigens:

Serology: C, c, E, e, K, k, $Fy^a$, $Fy^b$, $Jk^a$, $Jk^b$, M, S and s

Genotyping: C, c, E, e, VS, V, K, k, $Js^a$, $Js^b$, $Kp^a$, $Kp^b$, $Fy^a$, $Fy^b$, $Jk^a$, $Jk^b$, M, N, S, s, U, $Lu^a$, $Lu^b$, $Do^a$, $Do^b$, Hy, $Jo^a$, $LW^a$, $LW^b$, $Di^a$, $Di^b$, $Co^a$, $Co^b$, Sc1 and Sc2

Examples of the antigen combinations can be found in S1 Table where a full description of all 403 antigen combinations is provided as a supplement.

The current testing technology allows for red cell products to be categorised as one of 403 distinct antigen combinations using genotyping, and 299 distinct antigen combinations using extended phenotyping. For this study the following assumptions were used:

- donor panel consisted of 542,891 donors;

- approximately 700,000 donations collected annually;

- 13% of new donors did not return for a second donation;

- 17% of the panel is replaced with new donors each year;

- donors donate no more than 3 times per year;

- 12% of the donor panel has already been phenotyped;

- 670 donors are phenotyped each week;

- 135 donors are genotyped each week.

The probability for each antigen combination was determined and are provided in the supplement. Fig 2 displays the first 50 genotype antigen combinations with the highest probabilities (see S1 Table). The antigen combination with the highest frequency of being requested is c-E-K- with a probability of 0.41. Fig 2 demonstrates the rapid reduction in probabilities for the different genotype antigen combinations. Fig 3 zooms into the probabilities of the antigen

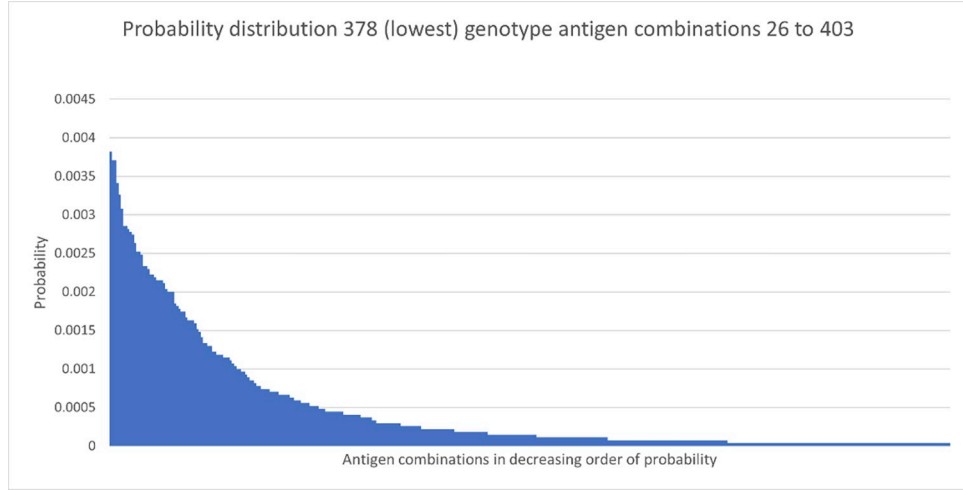

**Fig 3. Probabilities for genotype antigen combinations, with the first 25 combinations with the highest probabilities removed.** The horizontal axis shows antigen combinations 26 to 403 and the vertical axis shows probabilities.

**Table 3. Number of donors to be genotyped to achieve at least 95% and 85% success percentage of obtaining at least one donor of each antigen combination, based on the bound of Mallows.** Estimates given to 3 significant figures.

| Genotype estimates for donor panel of size 542,891 over 403 antigen combinations | | | | | |
|---|---|---|---|---|---|
| Minimum success percentage, C | $S_C$: Number of donors to achieve minimum percentage | Percentage donor panel | Lower bound | Upper bound | Range of bounds |
| 95% | 205,000 | 37.76% | 0.950001 | 0.951219 | 0.0012 |
| 85% | 175,300 | 32.29% | 0.850003 | 0.860614 | 0.0106 |

combinations with the first 25 removed, thus showing the continuing rate of decline across all combinations. It can be seen that approximately 93% of the antigen combinations have a frequency of less than 4 in 1000. It is acknowledged that, in practice, requests for antigen combinations, such as C-E-K- may be used to fill a request for say E-K-. However, amalgamating such antigen combinations would increase the probabilities, but not change the number of donors to be allocated to amalgamated antigen combinations. It is not expected that this will change the calculation of the number of donors needed to fulfil requests. In addition, some resolution information would be lost.

The analysis below will show that it is the large number of requests that are associated with antigen combinations with small probabilities that significantly increases the number of typed donors needed to fulfil the requests.

The bounds of Mallows in Theorem 2 were used to estimate that 205,000 donors need to be genotyped to obtain at least one donor for each of the 403 antigen combinations with 95% probability. This equates to 38% of the current donor panel. Since these lower and upper bounds differ by only 0.12% at the 95% success percentage, this indicates a high level of accuracy. A summary of the results is provided in Table 3.

With respect to phenotype testing, the number of distinct antigen combinations reduced to 299, with the probabilities recalibrated for the phenotype specific data. Using the same procedure based on Theorem 2, the number of donors to be phenotyped to obtain at least one donor for each of the 299 antigen combinations, with 95% probability, was estimated to be 35% of the donor panel or 188,000. A summary of the results is provided in Table 4. Once again there is a high degree of confidence in this estimate as the upper and lower bounds agree within 0.12%.

Validation of our bounds approach was conducted by performing Monte Carlo Multinomial sampling, as described previously. The results of this process are given in Figs 4 and 5. For Fig 4, we have assumed 23541 random donors are genotyped and three independent Monte Carlo sampling scenarios have been run. The figure, 23541, was estimated using the Mallows bound as in Eq (1), at the 95% success percentage and is based on the requirement that all targets are satisfied at least once. The target data were provided by the Red Cross for a 120-day time period. In this case, the value of each group of four bars represents one antigen combination and each colour represents a single assignment of donors to antigen combinations. The target requests are indicated by blue bars with the three Monte Carlo samplings

**Table 4. Number of donors to be phenotyped to achieve at least 95% and 85% success percentage of obtaining at least one donor of each antigen combination.** Estimates given to 4 significant figures.

| Phenotype estimates for donor panel of size 542,891 over 299 antigen combinations | | | | | |
|---|---|---|---|---|---|
| Minimum success percentage, C | $S_C$: Number of donors to achieve minimum success percentage | Percentage donor panel | Lower bound | Upper bound | Range of bounds |
| 95% | 188,000 | 34.60% | 0.950007 | 0.95121 | 0.00121 |
| 85% | 158,200 | 29.14% | 0.850012 | 0.86054 | 0.0105 |

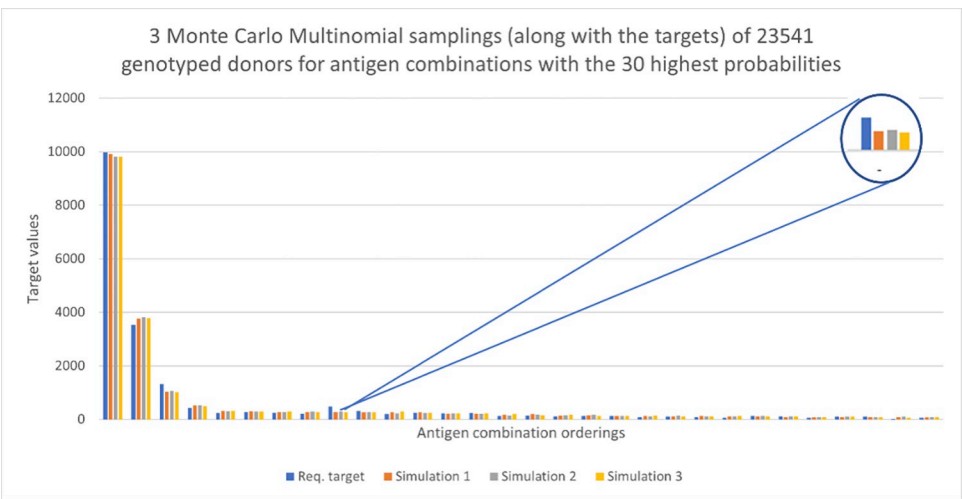

**Fig 4. Request targets for a 120-day period shown in blue, with three Monte Carlo Multinomial samplings.** Here there are 23,541 donors and we view the 30 antigen combinations with the highest probabilities.

shown in different colours. Here only the top 30 antigen combinations are displayed. There are approximately 10000 requests for the first antigen combination (c-E-K-) and the sampling process has allocated most of the donors to that antigen combination, but not enough to meet all the targets. It can also be seen that the target for the antigen combination denoted 9 has not been met (as highlighted in the insert in Fig 4). This is true for many of the other antigen combinations, while some targets have been exceeded.

In order to see how more targets can be met, we double the number of typed donors to $S = 47082$. We again run three independent Monte Carlo samplings. We can see in Fig 5 that many more targets are achieved for the 30 antigen combinations with the highest probabilities. We explore further the relationship between $S$ and $K$ (the total number of targets) in order to meet the requested targets later. Figs 6 and 7 show results for the 30 antigen combinations with the smallest probabilities, with again $S = 23541, 47082$, respectively. The value of $K$ for

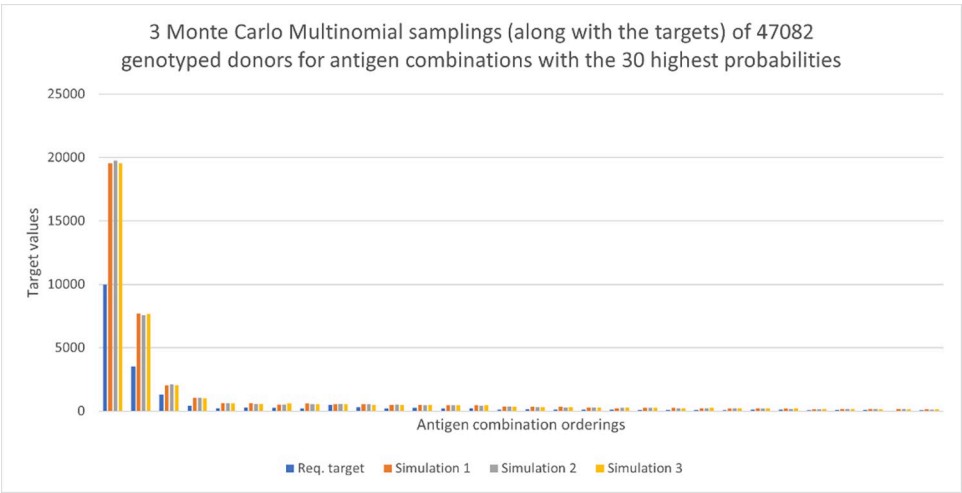

**Fig 5. Request targets for 120 days shown in blue, with three Monte Carlo Multinomial samplings.** Here there are 47,082 donors and we view the 30 antigen combinations with highest probabilities.

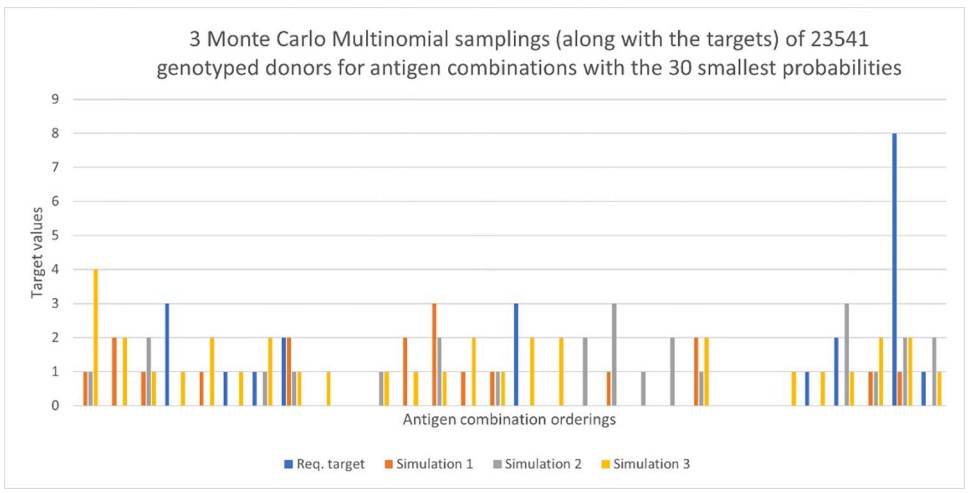

**Fig 6. Request targets for a 120-day period shown in blue, with three Monte Carlo Multinomial samplings.** Here there are 23,541 donors and we view the 30 antigen combinations with the smallest probabilities.

these two figures is 22. We see considerable variability across the three samplings in both cases, at these low levels of probability.

To investigate further the relationship between $S$ and $K$ at the 95% success percentage, a set of 10,000 Monte Carlo Multinomial samples were run. Targets for each of the 403 antigen combinations were determined for a given time period. The specific time periods that were chosen are:

- 15 consecutive 8-day periods (a possible preferred period for holding blood in inventory);

- 3 consecutive 120-day periods (assuming donors donate up to 3 times per year).

If $S = K$ then the probability of meeting all the targets $(k_1, \ldots, k_{403})$ is close to 0. However, as the number of genotyped donors increases, to $6^*K$, the probability of meeting targets is in excess of 94%, as shown in Table 5.

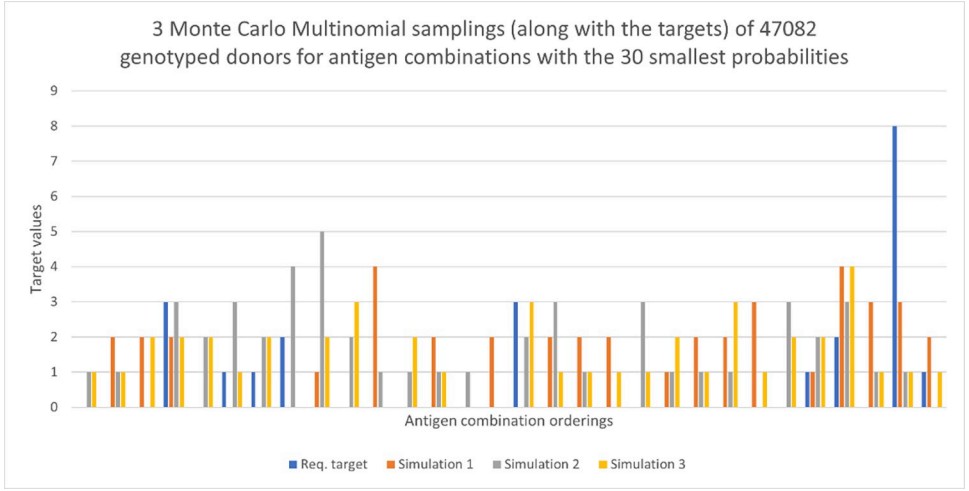

**Fig 7. Request targets for 120 days shown in blue, with three Monte Carlo Multinomial samplings.** Here there are 47,082 donors and we view the 30 antigen combinations with the smallest probabilities.

**Table 5. The probability of meeting targets for 403 antigen combinations based on fifteen 8-day periods and three 120-day periods, respectively, over 10,000 Monte Carlo Multinomial samplings.**

| Monte Carlo Multinomial sampling genotype | | | |
|---|---|---|---|
| Targets based on 8-day periods | | Targets based on 120-day periods | |
| S | Minimum success % | S | Minimum success % |
| 8600 | 92.80% | 94164 | 95.04% |
| 10750 | 94.04% | 117705 | 96.53% |
| 12900 | 94.54% | 141246 | 97.27% |
| 15050 | 95.04% | 164787 | 97.77% |
| 17200 | 95.53% | 188328 | 98.51% |

The slight disparity between these numbers with the estimate of genotyping 205,000 donors (Table 3) is to be expected as in a 120-day period not all rare antigen combinations will be requested. However the Monte Carlo Multinomial sampling must still allocate donors to these antigen combinations just in case they are needed.

The estimates provided in Table 4 have been validated with 10,000 Monte Carlo Multinomial samplings where the phenotyping request targets have been determined for 8 day and 120-day periods. The results are given in Table 6 where the number of categories is now 299. Once again results are consistent with the estimated number of donors to be phenotyped to achieve 95% success percentage as in Table 4.

For the purpose of illustrating the techniques presented here, the calculation of timelines for meeting genotyping levels are based on the following assumptions.

- Weekly donations are collected from 13,500 returning donors and 1% of these are genotyped per week.

- Currently $q = 13500 * 0.01 * 365/7 = 7,039$ donors genotyped annually.

- The current number of donors genotyped is $D_0 = 0.018 * 542891 = 9772$.

- $y$ represents the number of years of genotyping.

- $m$ represents the percentage of non-returning donors per annum, $m = 0.13$, i.e. 13%.

All calculations are again based on the formula

$$D_y = (1 - m)^y D_0 + \frac{M}{m} \left(1 - (1 - m)^y\right)$$

Table 7 shows the effect on the total number of donors genotyped for various typing percentages. Due to the assumed annual loss factor of 13% for non-returning donors, annual

**Table 6. The probability of meeting targets for 299 antigen combinations set from fifteen 8-day periods and three 120-day periods over 10,000 Monte Carlo Multinomial samplings.**

| Monte Carlo Multinomial sampling phenotype | | | |
|---|---|---|---|
| Targets set over 8-day periods | | Targets set over 120-day periods | |
| S | Minimum success Percentage | S | Minimum success Percentage |
| $4*K = 12888$ | 93.98% | $4*K = 94156$ | 95.32% |
| $5*K = 15036$ | 94.65% | $5*K = 117695$ | 95.99% |
| $6*K = 17184$ | 94.98% | $6*K = 141234$ | 97.32% |
| $7*K = 19332$ | 95.32% | $7*K = 164773$ | 97.66% |
| $8*K = 21480$ | 95.65% | $8*K = 188312$ | 98.66% |

**Table 7. Estimates (rounded to integers) of the number of years needed to reach 205,000 genotyped donors at the given percentages, minimum values that exceed target are shown.**

| | Given percentage of genotyping, the number of donors that will be genotyped after *y* years | | | | | | | | |
|---|---|---|---|---|---|---|---|---|---|
| | Genotyping percentage of donor panel | | | | | | | | |
| | ≤ 4% | 5% | 6% | 7% | 8% | 9% | 10% | 11% | 12% |
| Years to attain | Never attain | 12 | 8 | 7 | 5 | 5 | 4 | 4 | 3 |
| Number typed donors | N/A | 207381 | 207277 | 224388 | 208024 | 233419 | 221834 | 243458 | 213909 |

genotyping levels of 1% or even 4% are not enough to attain a cumulative 38% level for genotyping the donor panel within 20 years. However, if the level of genotyping is raised to 5% then 38% or 205,000 genotyped donors would be reached within 12 years, and if 12% of the donor panel is genotyped each year then 38% will be attained within 3 years.

Furthermore, once a level of 38% genotyping of the donor panel is reached and given that 13% of donors do not return, then an annual genotyping level of 2.14% of the donor panel will maintain this level at 38%.

The calculation of timelines for meeting phenotyping levels are based on the assumptions:

- Weekly donations are collected from 13,500 returning donors and 5% of these are phenotyped each week.

- Currently $q = 13500*0.05*365/7 = 35196$ donors are phenotyped each year.

- The current number of donors phenotyped is $D_0 = 542,891*0.121 = 65689$.

- $y$ represents the number of years of phenotyping.

- $m$ represents the percentage of non-returning donors per annum, $m = 0.13$, 13%.

Table 8 gives estimates of the effect of increasing the percentage of phenotyping. If the level of phenotyping is retained at 5% of the donor panel, then 35% of the donor panel (188,000 donors) will be phenotyped within 8 years and if the level of phenotyping is increased to 9% of the donor panel, then a 35% target will be reached in 3 years. Furthermore, once a level of 35% phenotyping of the donor panel is reached and given that 13% of donors are not returning, then an annual phenotyping level of 1.94% of the donor panel will maintain this level at 35%.

## Conclusions

We have used a Multinomial distribution to provide estimates for how many donors should be tested to meet Lifeblood's requests with respect to genotyping and phenotyping. The model was constructed based on recent historical blood request data for uncommon red cell antigen combinations, supplied by Lifeblood. An important mathematical technique is based on the upper and lower bounds of the multinomial probability of attaining a set of bounds as given by Mallows in Theorem 2. In doing this analysis we have assumed that the demand is uniformly distributed over time. Over long time periods this is an appropriate assumption.

**Table 8. Estimates (rounded to integers) of the number of years needed to reach 188,000 phenotyped donors at the given percentages, minimum values that exceed target are shown.**

| | Given percentage of phenotyping, the number of donors that will be phenotyped after *y* years | | | | | |
|---|---|---|---|---|---|---|
| | Phenotyping percentage of donor panel | | | | | |
| | ≤ 4% | 5% | 6% | 7% | 8% | 9% |
| Years to attainment | Never attained | 8 | 6 | 4 | 4 | 3 |
| Number typed donors | N/A | 191619 | 200533 | 188999 | 210622 | 198862 |

Genotyping of 38% of the donor panel is required to meet demand with at least a 95% success percentage. At the current level of 1% genotyping of weekly inventory, it is estimated that it is not possible to reach a target of 38% (205,000 donors). If the weekly genotyping levels are raised to 5% the target will be reached within 12 years. Increasing to 12% would allow the 38% target to be reached within 3 years. The model demonstrated that when the target is reached then annual genotyping of an additional 28,500 is required to maintain the overall level.

Phenotyping of 35% of the donor panel is required to meet Lifeblood's demand with at least a 95% success percentage. At the current level of 5% phenotyping of inventory, it is estimated that it will take 8 years to reach a target of 35% (188,000). If the level of extended phenotyping of weekly returning donors is raised to 9% then a target would be achieved within 3 years. The model also demonstrated that when the target is reached then annual extended phenotyping of an additional 26,140 is required to maintain the overall level.

With the given mathematical model, we have the above access to tight lower and upper bounds on the multinomial probabilities of meeting a set of targets, due to the theory of Mallows. This makes rare event estimations much more efficient than just through Monte Carlo simulations. This is an undoubted strength of our work. We have assumed that demand is uniformly distributed over time and not adjusted for that. However, over long time periods this is not an unreasonable assumption. Another possible limitation is access to further data. We were able to use request data for 2015/2016 to estimate antigen combination frequencies. However, our estimations could have been more effective if request data for later years had been used for further comparison and validation of results. However, we did not have access to additional data. The result is a mathematical model that will inform business decisions and assist Lifeblood to determine the level of investment required to meet the desired timeline to achieve the optimum panel size.

## Supporting information

**S1 Table. Antigen combinations that are met only by extended phenotyping.**
(DOCX)

## Acknowledgments

Australian governments fund Australian Red Cross Lifeblood for the provision of blood, blood products and services to the Australian community.

## Author Contributions

**Conceptualization:** Tanya Powley, James Daly.

**Data curation:** Kevin Burrage, Pamela Burrage, Diane Donovan, Bevan Thompson.

**Formal analysis:** Kevin Burrage, Pamela Burrage, Diane Donovan, Bevan Thompson.

**Investigation:** Tanya Powley.

**Project administration:** Denisse Best, James Daly.

**Resources:** Tanya Powley.

**Supervision:** James Daly.

**Writing – original draft:** Denisse Best, Kevin Burrage, Pamela Burrage, Diane Donovan, Shamila Ginige, Bevan Thompson.

**Writing – review & editing:** Denisse Best, Shamila Ginige, James Daly.

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
