## [Decision Letter · Decision Letter 0]

20 Dec 2021

PONE-D-21-20968Probabilistic mathematical modelling to predict the red cell phenotyped donor panel sizePLOS ONE

Dear Dr. Ginige,

Thank you for submitting your manuscript to PLOS ONE. After careful consideration, we feel that it has merit but does not fully meet PLOS ONE’s publication criteria as it currently stands. Therefore, we invite you to submit a revised version of the manuscript that addresses the points raised during the review process.

Out of respect of timelyness, or lack thereof, I am acting on 1 review.  Please address the reviewers comments. 

We look forward to receiving your revised manuscript.

Kind regards,

Jeffrey Chalmers, Ph.D.

Academic Editor

PLOS ONE

Journal Requirements:

3. We note you have included a table to which you do not refer in the text of your manuscript. Please ensure that you refer to Table 10 and 11 in your text; if accepted, production will need this reference to link the reader to the Table.

Reviewers' comments:

Reviewer's Responses to Questions

**Comments to the Author**

1. Is the manuscript technically sound, and do the data support the conclusions?

Reviewer #1: Yes

2. Has the statistical analysis been performed appropriately and rigorously? 

Reviewer #1: Yes

3. Have the authors made all data underlying the findings in their manuscript fully available?

Reviewer #1: Yes

4. Is the manuscript presented in an intelligible fashion and written in standard English?

Reviewer #1: Yes

5. Review Comments to the Author

Reviewer #1: I would like to thank the authors for their work here. I think it's quite interesting and covers an area not well reported in the literature.

Please see the attached PDF for a marked up version of the paper. You may wish to use Adobe reader to view the marked up file.

Some comments:

Ln 121: I think, in fairness, that the paper by Blake and Clarke was primarily designed to evaluate the impact of frozen inventory.

Ln 129: Well, the mulitnomial distribution is just a generalization of the binomial distribution. I don't think independence vs. not independent is correct here. If we had 3 categories (X, Y, or Other) and compressed that to 2 (X and Other), we are still modelling the same selection process, we just have different categories for observation. Categories are mutually exclusive and if the number of categories is 2 we call it binomial and if 3 or more, it's multinomial. Same process, no difference in the idea of independence.

Ln 173: I understand the algorithm, of course, but a single donor is unlikely to be of great value, since the average transfusion request is typically greater than a single unit. So, the question then becomes one of having sufficient donors available (or units frozen) to meet an average demand with more availability being better.

Ln 238: Why m/2? Obviously this could be any fraction one likes, but why assume m/2? In particular, it appears later in the paper that 17% of donors don't return and 13% of new donors don't return. Could the authors comment and perhaps reconcile?

Figures 3 and 4: Could labels be added to axes of the graphs?

Ln 325: Why 23541? The rationale for this number doesn't become apparent until much later.

Ln 334: There's a lot of variance, one would assume in the results. Why is an n of 3 sufficient to draw conclusions?

Figure 7: Data in Figure 7 is hard to decipher. Take antigen combination 3 in the chart. No target is shown (is it 0?). Consider also combination 4. Is there a target, but no simulation results? And also, when the authors say that the figure shows that a 95% CI is validated, how does one see this in the figure? Could a different method of displaying the data be considered?

Ln 370: It's confusing to talk about 10,000 simulations and the graph to show 3. Could this be changed to say 3 replications of 10,000 samples?

Ln 375: Of course, this is one method for testing for sufficiency, but demand is likely to be less uniformly distributed over time than assumed.

Table 5: Perhaps show values of K = 4 through 6 or 7 so that readers can get a sense of the improvement on the way to 95% rather than once exceeded.? Similar comments for 120 days.

6. PLOS authors have the option to publish the peer review history of their article (what does this mean?). If published, this will include your full peer review and any attached files.

Reviewer #1: No

---

## [Author Response · Author response to Decision Letter 0]

3 Feb 2022

Dear Prof. Chalmers,

Thank you for the opportunity to resubmit the manuscript “Probabilistic mathematical modelling to predict the red cell phenotype donor panel size”. We would also like to thank the reviewer for his/her careful reading of the article and the insightful comments that we have addressed in full and have improved the presentation. Please see below for details where we have responded to the requirements of the journal and all the reviewer’s comments.

Journal Requirements: 

The aforementioned templates have been reviewed and the manuscript meets the requirements mentioned in them.

The data set for this study is given in S1 Table in the appendix and there is no other data associated with this study. This data has been uploaded to the supplementary material file.

3. We note you have included a table to which you do not refer in the text of your manuscript. Please ensure that you refer to Table 10 and 11 in your text; if accepted, production will need this reference to link the reader to the Table.

Thank you for pointing this out. This material was additional analysis of the material given in the S1 Table and as such can be recovered from that table. Since we do not directly refer to these tables in the manuscript we have removed them. 

Response to Reviewer's Comments/Questions

5. Review Comments to the Author

Ln 121: I think, in fairness, that the paper by Blake and Clarke was primarily designed to evaluate the impact of frozen inventory.

This has been noted in the text.

Ln 129: Well, the multinomial distribution is just a generalization of the binomial distribution. I don't think independence vs. not independent is correct here. If we had 3 categories (X, Y, or Other) and compressed that to 2 (X and Other), we are still modelling the same selection process, we just have different categories for observation. Categories are mutually exclusive and if the number of categories is 2 we call it binomial and if 3 or more, it's multinomial. Same process, no difference in the idea of independence.

Thanks. The text has been reworded here in line with comments.

Ln 173: I understand the algorithm, of course, but a single donor is unlikely to be of great value, since the average transfusion request is typically greater than a single unit. So, the question then becomes one of having sufficient donors available (or units frozen) to meet an average demand with more availability being better.

Additional text has been added here to clarify the intent of the text.

Ln 238: Why m/2? Obviously this could be any fraction one likes, but why assume m/2? In particular, it appears later in the paper that 17% of donors don't return and 13% of new donors don't return. Could the authors comment and perhaps reconcile?

Additional text has been added to clarify the choice of m/2.

Figures 3 and 4: Could labels be added to axes of the graphs?

Labels have been added as requested.

Ln 325: Why 23541? The rationale for this number doesn't become apparent until much later.

An explanation of this has been added to the text.

Ln 334: There's a lot of variance, one would assume in the results. Why is an n of 3 sufficient to draw conclusions?

The number of samples used was 10,000 however only three were displayed in the figures to highlight the variability that the Reviewer quite rightly points out.

Figure 7: Data in Figure 7 is hard to decipher. Take antigen combination 3 in the chart. No target is shown (is it 0?). Consider also combination 4. Is there a target, but no simulation results? And also, when the authors say that the figure shows that a 95% CI is validated, how does one see this in the figure? Could a different method of displaying the data be considered?

Figures 4, 5, 6 and 7 have been reordered and edited reducing the number of antigen combinations to improve readability, while still conveying the same information. 

Ln 370: It's confusing to talk about 10,000 simulations and the graph to show 3. Could this be changed to say 3 replications of 10,000 samples?

The text has been reworded to clarify the stated information.

Ln 375: Of course, this is one method for testing for sufficiency, but demand is likely to be less uniformly distributed over time than assumed.

In validating the earlier results different scenarios for Monte Carlo sampling have been considered. Simulations have been run on the entire years data, simulations have been run on 15 non overlapping 8 day periods and 3 non overlapping 120 day periods, where 8 days and 120 days periods have been chosen based on LifeBlood procedures with respect to the storage of blood products and the returning donors. Since the 120 day simulation results were in line with all other results 3 consecutive periods was deemed to be enough. Where necessary donors can be approached to give blood 4 times a year.

Table 5: Perhaps show values of K = 4 through 6 or 7 so that readers can get a sense of the improvement on the way to 95% rather than once exceeded.? Similar comments for 120 days.

The tables have been recast to meet the Reviewers request.

---

## [Decision Letter · Decision Letter 1]

17 Mar 2022

PONE-D-21-20968R1Probabilistic mathematical modelling to predict the red cell phenotyped donor panel sizePLOS ONE

Dear Dr. Ginige,

Thank you for submitting your manuscript to PLOS ONE. After careful consideration, we feel that it has merit but does not fully meet PLOS ONE’s publication criteria as it currently stands. Therefore, we invite you to submit a revised version of the manuscript that addresses the points raised during the review process.

While my decision is based on just one review, this reviewer has significant questions, and suggests another round of reviews.  To keep the process moving, I agree..

We look forward to receiving your revised manuscript.

Kind regards,

Jeffrey Chalmers, Ph.D.

Academic Editor

PLOS ONE

Reviewers' comments:

Reviewer's Responses to Questions

**Comments to the Author**

1. If the authors have adequately addressed your comments raised in a previous round of review and you feel that this manuscript is now acceptable for publication, you may indicate that here to bypass the “Comments to the Author” section, enter your conflict of interest statement in the “Confidential to Editor” section, and submit your "Accept" recommendation.

Reviewer #1: (No Response)

2. Is the manuscript technically sound, and do the data support the conclusions?

Reviewer #1: Yes

3. Has the statistical analysis been performed appropriately and rigorously? 

Reviewer #1: Yes

4. Have the authors made all data underlying the findings in their manuscript fully available?

Reviewer #1: Yes

5. Is the manuscript presented in an intelligible fashion and written in standard English?

Reviewer #1: Yes

6. Review Comments to the Author

Reviewer #1: I'm sorry, but I don't seem to have access to line numbers in this version of the manuscript and the page numbers are not printed on the manuscript, so I cannot be very specific about the location of items that need to be addressed. I will do my best to be helpful here, though.

Pg 11: compatible RBC  compatible RBCs

Pg 15: blood donor phenotyped  blood donors phenotyped

Pg 18: Sentence starting "Of course, the one donor..." Honestly, I don't understand this sentence. Could it be reworded or expanded?

Pg 18: I apologize - I probably should have asked this question in the first review. I don't understand what the authors mean when they say they say "the allocation of to a category is achieved by a random process based on a Multinomial distribution where S donors are allocated across r categories based on the category probabilities in Fig 2". Can the authors give more detail on how donors are allocated to categories? I'm struggling to think about how this would be implemented as Multinomial draw (which would require an enumeration of all possible values of n1, n2, ..., nr adding up to S). Is it a draw from a cumulative distribution for a donor that adds them to a category? Please more detail here, since it would be hard for others to reproduce the method without understanding this crucial step.

Pg 20, 1st Equation: Can you perhaps add more space between E[ni] and Var[ni]? The two look like a single equation.

Pg 20, 2nd Equation: add a space between for and j

Pg 21, Equation under Theorem 2: What does the 2nd equation mean? Is it missing Ps? The first line appears (I didn't check the reference to Mallow) to set U and L bounds on Ps. So, what is the 2nd equation? Please clarify.

Pg 21 "within the given time periods". No mention of time periods has appeared in the paper to this point. You need to give the readers a "heads up" about this.

Pg 23: Sometimes called timelines, sometimes called timeframes. Please be consistent.

Pg 23: Also, it's not clear to readers why timelines are introduced. Please give a sentence or two.

Pg 24: The authors added a sentence about the selection of m/2 in the 1st equation on the page, but it doesn't answer my question. If m is the retention rate for donors, why is the retention rate m/2 for new donors? Do you mean that they are, on average, only in the system for 1/2 year? As it's written this says there is a different retention rate for new donors and that it's less than repeat donors. This is somewhat contradictory to most literature that says new donors have very poor retention rates. Regardless of the value used (m, m/2 or something different), the authors need to tell the readers why the value is an appropriate model for new donor retention.

Pg 33: Tables 5 and 6 have exactly the same titles. Please distinguish what's appearing in the table.

7. PLOS authors have the option to publish the peer review history of their article (what does this mean?). If published, this will include your full peer review and any attached files.

Reviewer #1: No

---

## [Author Response · Author response to Decision Letter 1]

27 Apr 2022

RESPONSE TO REVIEWER’S REPORT

Again, we would like to thank the Reviewer for their careful reading of the manuscript and their insightful comments which have allowed us to improve our presentation. 

We have addressed all comments made Reviewer and made changes as outlined below.

Pg 11: compatible RBC  compatible RBCs

Changed as required.

Pg 15: blood donor phenotyped  blood donors phenotyped

Changed as required.

Pg 18: Sentence starting "Of course, the one donor..." Honestly, I don't understand this sentence. Could it be reworded or expanded?

The wording here has been changed as well taking into account the reviewers comments. The given sentence refers to the calculations within the simulation which is a modelled using a Mutlinomial distribution sampling

Pg 18: I apologize - I probably should have asked this question in the first review. I don't understand what the authors mean when they say they say "the allocation of to a category is achieved by a random process based on a Multinomial distribution where S donors are allocated across r categories based on the category probabilities in Fig 2". Can the authors give more detail on how donors are allocated to categories? I'm struggling to think about how this would be implemented as Multinomial draw (which would require an enumeration of all possible values of n1, n2, ..., nr adding up to S). Is it a draw from a cumulative distribution for a donor that adds them to a category? Please more detail here, since it would be hard for others to reproduce the method without understanding this crucial step.

Additional text has been added to clarify this situation.

In general terms this type of allocation problem arises frequently enough that a number of programming languages, including Python, Matlab and R provide implementations often based on acceptance/rejection sampling. For instance in Matlab it is implemented in the command mnrnd(S,p) where p is the vector of probabilities of categories with length r, the number of categories, and S is the number of donations.

Pg 20, 1st Equation: Can you perhaps add more space between E[ni] and Var[ni]? The two look like a single equation.

The equations have now been split across 2 lines. 

Pg 20, 2nd Equation: add a space between for and j

Pg 21, Equation under Theorem 2: What does the 2nd equation mean? Is it missing Ps? The first line appears (I didn't check the reference to Mallow) to set U and L bounds on Ps. So, what is the 2nd equation? Please clarify.

We thank the referee for drawing our attention to this. We have taken the statement out of the equation and clarified the value of S95 

Pg 21 "within the given time periods". No mention of time periods has appeared in the paper to this point. You need to give the readers a "heads up" about this.

The reference to time periods is not necessary here and will be dealt with later when they arise. So the statement has been removed.

Pg 23: Sometimes called timelines, sometimes called timeframes. Please be consistent.

We have changed all references to ‘timelines’

Pg 23: Also, it's not clear to readers why timelines are introduced. Please give a sentence or two.

The development of a timeline refers to the development of a testing schedule to meet request targets. Additional information has been given to make this clearer.

Pg 24: The authors added a sentence about the selection of m/2 in the 1st equation on the page, but it doesn't answer my question. If m is the retention rate for donors, why is the retention rate m/2 for new donors? Do you mean that they are, on average, only in the system for 1/2 year? As it's written this says there is a different retention rate for new donors and that it's less than repeat donors. This is somewhat contradictory to most literature that says new donors have very poor retention rates. Regardless of the value used (m, m/2 or something different), the authors need to tell the readers why the value is an appropriate model for new donor retention.

We thank the reviewer for drawing our attention to this. The calculations are consistent, but the wording has been improved to better reflect the underlying mathematics. Essentially, m=0.13 and, for example, there are around 35000 new donors phenotyped, a number that is currently increasing, and around 65600 already phenotyped. Using m/2 with M=q(1-m/2) gives the average number of donors donating during the year and thus the average supply

Pg 33: Tables 5 and 6 have exactly the same titles. Please distinguish what's appearing in the table.

The captions for Tables 5 and 6 have been updated to accurately distinguish between them.

7. PLOS authors have the option to publish the peer review history of their article (what does this mean?). If published, this will include your full peer review and any attached files.

This will not be necessary.

---

## [Decision Letter · Decision Letter 2]

13 Jul 2022

PONE-D-21-20968R2Probabilistic mathematical modelling to predict the red cell phenotyped donor panel sizePLOS ONE

Dear Dr. Ginige,

Thank you for submitting your manuscript to PLOS ONE. After careful consideration, we feel that it has merit but does not fully meet PLOS ONE’s publication criteria as it currently stands. Therefore, we invite you to submit a revised version of the manuscript that addresses the points raised during the review process.

I was not able to secure reviews from the reviewers for the first version of the mansucript, and as you can see, one of the reviewers had significant concerns.  I will give you a chance to address those concerns..

We look forward to receiving your revised manuscript.

Kind regards,

Jeffrey Chalmers, Ph.D.

Academic Editor

PLOS ONE

Reviewers' comments:

Reviewer's Responses to Questions

**Comments to the Author**

1. If the authors have adequately addressed your comments raised in a previous round of review and you feel that this manuscript is now acceptable for publication, you may indicate that here to bypass the “Comments to the Author” section, enter your conflict of interest statement in the “Confidential to Editor” section, and submit your "Accept" recommendation.

Reviewer #1: (No Response)

Reviewer #2: (No Response)

2. Is the manuscript technically sound, and do the data support the conclusions?

Reviewer #1: Yes

Reviewer #2: Partly

3. Has the statistical analysis been performed appropriately and rigorously? 

Reviewer #1: Yes

Reviewer #2: Yes

4. Have the authors made all data underlying the findings in their manuscript fully available?

Reviewer #1: Yes

Reviewer #2: Yes

5. Is the manuscript presented in an intelligible fashion and written in standard English?

Reviewer #1: Yes

Reviewer #2: Yes

6. Review Comments to the Author

Reviewer #1: Thank you for the revisions to the paper and the detailed work to address the items in R1.

I hate to do this, but I still don't understand the choice of m/2 as the retention rate for newly screened donors. The actual value (m/2) isn't important to the derivation that follows (it's part of the parameter M), but the selection of m/2 is not explained in the text.

To make matters worse, the explanation in the response to reviewers doesn't make sense to me and differs from what's written in the text.

The equation is:

D1 = (1-m)D0 + q(1-m/2)

Where:

Dy is the number of donors in the system in year y

q is the fraction of non-returning screened donors per annum as a decimal

Ok, so fine. The number of donors in D1 = Donors at D0 x retention rate + Screened donors x (a different retention rate)

It's not obvious why the non-screened donors have a greater likelihood of being retained than already screened donors.

The text says that m/2 is chosen as the mean of a uniform distribution of donors throughout the year. However, the note to reviewers says m = 0.13 and says "using m/2 gives the average number of donors donating during the year and thus the average supply".

Neither of these two explanations really is clear. I think what's meant here is that there are q donors screened over the course of the year and thus, on average, a newly screened donor has been registered for six months at the end of the first year. Accordingly, the non-return rate has been pro-rated by m/2 to reflect this fact.

Could the wording perhaps be changed to make this clear? Otherwise the selection of m/2 looks arbitrary. Given that it disappears in the following line into M, the reader is left wondering why the parameter was changed.

Finally, if an empirical value of m can be found (m = 0.13), then it would seem to me that an empirical value of m' for new donors could be found from data and the assumption of m/2 avoided (or perhaps confirmed).

Reviewer #2: The authors present results from a probabilistic model to meet requests for specific antigen combinations in Australia. The authors use data from Lifeblood (the sole provider for fresh blood components to Australians) to understand distributions of requests across antigen combinations and specific targets over time periods. The authors are able to estimate total donor sizes needed to fulfill requests with probability 0.95 and to estimate the length of time (and percentage of geno- or pheno-typing) it would take to achieve the necessary number of geno- (or pheno-) typed samples. The manuscript will be strengthened if the authors address the following points.

1. In the Abstract (line 12-14), authors state that simulations were used to attain a 95% confidence interval, but that doesn't seem to be the case. Simulations appear to have been run to estimate a minimum success percentage. No confidence intervals are presented.

2. On page 10, lines 1-8 (describing Figure 2), I am a bit confused. Figure 2 appears to show the distribution of the requests for certain genotype antigen combinations based on historical Lifeblood data. These probabilities then seem to be used as probabilities that donors will have a particular genotype-antigen combination. This may be the best the authors can do, but I think it warrants a statement possibly in the methods, but also likely as a limitation in the Conclusions section, that request probabilities may not be representative of the distribution of donors. If I am misunderstanding something, authors should clarify.

3. On pages 12 and13, the notation is a bit confusing, since authors start using n_j as a random variable, by giving its expected value and variance and using it in the probability expression on line 4 of page 13. However, on page 11, authors use n_j to refer to the "observed" number of requests (used in the multinomial probability expression). Although minor, authors might consider distinguishing between the random variable and the observed value (often, in probability, capital letters refer to a random variable and lower case letters refer to the observed value).

4. Also on page 13, line 5, authors state an assumption that n_j >= k_j. The probability in the previous line is for the probability that all n_i>=k_i, so if that is assumed, then the probability calculation isn't all that interesting. I don't believe authors need to state n_j>= k_j in line 5.

5. In Theorem 2, do authors really need the 2nd bound given in line 17? As written, it is not all that interesting (I actually think there is something incorrect here since Mallow's paper has something different, even when translated to the direction of inequalities used here). As written, the smallest line 17 can be is 1 and the largest it can be is e^r. We already know that probabilities have to be less than or equal to 1, so it really doesn't add any information.

6. The reference for the proof to Theorem 2, I think should be reference #11 not #10

7. In response to another reviewer, authors added a definition of S_95 on line 10, page 14, but I actually found this a bit confusing, since it suggests authors estimate the n_j. Assuming, I am understanding the steps authors took to estimate S_95, authors only need the p_j (which are from Figure 2, supplemental table), k_i (which I'm assuming are also from the supplemental table) and then they can perform the probability calculations in the lower bound at different S to find the S that gets a lower bound greater than 0.95. If the steps are actually different, authors should clarify what is done (they may want to clarify even if what I describe is correct, since not everyone will be able to follow what is currently written).

8. On page 15, lines 1-5 are a bit confusing, especially the calculation of the minimum percentage of success in attaining the targets. How can the number of failures to meet targets be calculated as a percentage of the number of categories r (unless this is done for each of the 10000 samples)? Is a failure at the sample level (so any sample that does not meet targets) or at the category level across the samples, so the total number of categories across the 10000 samples for which the targets were not met? Some clarification is needed.

9. Authors use "confidence level" in a lot of different places (including line 7, page 15, Tables 3 and 4 and related text), but this term is used specifically in statistics for something other than what it seems the authors are doing (unless authors are calculating minimum percentage of success differently than stated). Authors, in some cases (such as Tables 3 and 4) are really just talking about a probability (or lower bound of a probability).

10. The added explanation for m/2 in line 5 page 16 is still not clear. Why does "m" impact the number of new donors? Authors say that m/2 is from the Uniform, which would suggest they are working with a Uniform distribution on (0, m), but it isn't clear why that would make sense for number of donors. Authors need to provide more explanation.

11. lines 2 and 11 on page 20 (and titles to Tables 3 and 4): this may be related to my point #7 above, but authors refer to "to obtain at least one donor for each...". That would suggest that k_i in the equation in line 9, page 14, is equal to 1 for all i. Is that really what is being done?

12. When authors refer to 3 sampling scenarios in Figures 4-7, those are just 3 different multinomial samples from a particular multinomial distribution right? They aren't different scenarios...

13. page 22, lines 2-4: what is meant by the number being estimated by Mallows' bounds (again, this could be due to my confusion in point 7)? Is something different done here to get 23541 (which I assume is what was estimated) maybe based on different targets than what were used in calculations for Table 3?

14. page 22, line 4: "The target data was" should be "The target data were"

15. Authors should clarify the time periods used for Tables 5 and 6. In particular, I'm a bit confused how the consecutive time periods come in - is there something different in saying targets are based on 15 consecutive 8-day periods versus targets were based on 144 consecutive days of requests, for example, or was there something distinct about the 8-day (or 120-day) periods that were somehow used?

16. Table 5: Why is the first entry for S=0 under "Targets based on 8-day periods"?

17. It is not clear why authors feel that the multinomial simulations validate the numbers calculated using the Mallows' bound, especially for phenotyping, where Table 4 suggests 188,000 donors need to be phenotyped, while Table 6 has 95% probability at about half that size (94,156). Yes, the Mallows' bound value will exceed 95% probability based on the simulations, but that is potentially quite a bit more phenotyping than would be needed based on the simulations.

18. Authors should provide some discussion of strengths and limitations in their conclusions.

7. PLOS authors have the option to publish the peer review history of their article (what does this mean?). If published, this will include your full peer review and any attached files.

Reviewer #1: No

Reviewer #2: No

---

## [Author Response · Author response to Decision Letter 2]

16 Aug 2022

Reviewer #1: Thank you for the revisions to the paper and the detailed work to address the items in R1.

I hate to do this, but I still don't understand the choice of m/2 as the retention rate for newly screened donors. The actual value (m/2) isn't important to the derivation that follows (it's part of the parameter M), but the selection of m/2 is not explained in the text.

To make matters worse, the explanation in the response to reviewers doesn't make sense to me and differs from what's written in the text.

The equation is:

D1 = (1-m)D0 + q(1-m/2)

Where:

Dy is the number of donors in the system in year y

q is the fraction of non-returning screened donors per annum as a decimal

Ok, so fine. The number of donors in D1 = Donors at D0 x retention rate + Screened donors x (a different retention rate)

It's not obvious why the non-screened donors have a greater likelihood of being retained than already screened donors.

The text says that m/2 is chosen as the mean of a uniform distribution of donors throughout the year. However, the note to reviewers says m = 0.13 and says "using m/2 gives the average number of donors donating during the year and thus the average supply".

Neither of these two explanations really is clear. I think what's meant here is that there are q donors screened over the course of the year and thus, on average, a newly screened donor has been registered for six months at the end of the first year. Accordingly, the non-return rate has been pro-rated by m/2 to reflect this fact.

Could the wording perhaps be changed to make this clear? Otherwise the selection of m/2 looks arbitrary. Given that it disappears in the following line into M, the reader is left wondering why the parameter was changed.

Finally, if an empirical value of m can be found (m = 0.13), then it would seem to me that an empirical value of m' for new donors could be found from data and the assumption of m/2 avoided (or perhaps confirmed).

We thank Reviewer 1 for his comments on the calculations involving m/2 and have inserted the suggested wording into the text. 

Reviewer #2: The authors present results from a probabilistic model to meet requests for specific antigen combinations in Australia. The authors use data from Lifeblood (the sole provider for fresh blood components to Australians) to understand distributions of requests across antigen combinations and specific targets over time periods. The authors are able to estimate total donor sizes needed to fulfill requests with probability 0.95 and to estimate the length of time (and percentage of geno- or pheno-typing) it would take to achieve the necessary number of geno- (or pheno-) typed samples. The manuscript will be strengthened if the authors address the following points.

1. In the Abstract (line 12-14), authors state that simulations were used to attain a 95% confidence interval, but that doesn't seem to be the case. Simulations appear to have been run to estimate a minimum success percentage. No confidence intervals are presented.

We thank the Reviewer for this comment and have appropriately changed all instances to ``success percentage’’.

2. On page 10, lines 1-8 (describing Figure 2), I am a bit confused. Figure 2 appears to show the distribution of the requests for certain genotype antigen combinations based on historical Lifeblood data. These probabilities then seem to be used as probabilities that donors will have a particular genotype-antigen combination. This may be the best the authors can do, but I think it warrants a statement possibly in the methods, but also likely as a limitation in the Conclusions section, that request probabilities may not be representative of the distribution of donors. If I am misunderstanding something, authors should clarify.

This point has been addressed when we address point 18 and have included information on the limitations and strengths of the model.

3. On pages 12 and13, the notation is a bit confusing, since authors start using n_j as a random variable, by giving its expected value and variance and using it in the probability expression on line 4 of page 13. However, on page 11, authors use n_j to refer to the "observed" number of requests (used in the multinomial probability expression). Although minor, authors might consider distinguishing between the random variable and the observed value (often, in probability, capital letters refer to a random variable and lower-case letters refer to the observed value).

We acknowledge the Reviewers comments, however we feel that given the context the meaning is clear enough and prefer not to change it at this late stage.

4. Also on page 13, line 5, authors state an assumption that n_j >= k_j. The probability in the previous line is for the probability that all n_i>=k_i, so if that is assumed, then the probability calculation isn't all that interesting. I don't believe authors need to state n_j>= k_j in line 5.

As suggested by the Reviewer this information has been removed.

5. In Theorem 2, do authors really need the 2nd bound given in line 17? As written, it is not all that interesting (I actually think there is something incorrect here since Mallow's paper has something different, even when translated to the direction of inequalities used here). As written, the smallest line 17 can be is 1 and the largest it can be is e^r. We already know that probabilities have to be less than or equal to 1, so it really doesn't add any information.

When the lower bound is greater than or equal to 0.95, then the difference between the upper and lower bound is extremely small, so that the lower bound is a good estimate of the probability and can be calculated directly.

6. The reference for the proof to Theorem 2, I think should be reference #11 not #10

Corrected as given.

7. In response to another reviewer, authors added a definition of S_95 on line 10, page 14, but I actually found this a bit confusing, since it suggests authors estimate the n_j. Assuming, I am understanding the steps authors took to estimate S_95, authors only need the p_j (which are from Figure 2, supplemental table), k_i (which I'm assuming are also from the supplemental table) and then they can perform the probability calculations in the lower bound at different S to find the S that gets a lower bound greater than 0.95. If the steps are actually different, authors should clarify what is done (they may want to clarify even if what I describe is correct, since not everyone will be able to follow what is currently written).

We thank the Reviewer for this comment and have removed the summation. 

8. On page 15, lines 1-5 are a bit confusing, especially the calculation of the minimum percentage of success in attaining the targets. How can the number of failures to meet targets be calculated as a percentage of the number of categories r (unless this is done for each of the 10000 samples)? Is a failure at the sample level (so any sample that does not meet targets) or at the category level across the samples, so the total number of categories across the 10000 samples for which the targets were not met? Some clarification is needed.

Addtional wording has been given to clarify this point.

9. Authors use "confidence level" in a lot of different places (including line 7, page 15, Tables 3 and 4 and related text), but this term is used specifically in statistics for something other than what it seems the authors are doing (unless authors are calculating minimum percentage of success differently than stated). Authors, in some cases (such as Tables 3 and 4) are really just talking about a probability (or lower bound of a probability).

This has been addressed as in point 1 above.

10. The added explanation for m/2 in line 5 page 16 is still not clear. Why does "m" impact the number of new donors? Authors say that m/2 is from the Uniform, which would suggest they are working with a Uniform distribution on (0, m), but it isn't clear why that would make sense for number of donors. Authors need to provide more explanation.

This has been addressed as stated in the response to Reviewer 1.

11. lines 2 and 11 on page 20 (and titles to Tables 3 and 4): this may be related to my point #7 above, but authors refer to "to obtain at least one donor for each...". That would suggest that k_i in the equation in line 9, page 14, is equal to 1 for all i. Is that really what is being done?

Yes, this is correct as stated in a number of places in the article.

12. When authors refer to 3 sampling scenarios in Figures 4-7, those are just 3 different multinomial samples from a particular multinomial distribution right? They aren't different scenarios...

Three specific different instances of request data were used to set data and to validate the theoretical results. So, three different samples have been used.

13. page 22, lines 2-4: what is meant by the number being estimated by Mallows' bounds (again, this could be due to my confusion in point 7)? Is something different done here to get 23541 (which I assume is what was estimated) maybe based on different targets than what were used in calculations for Table 3?

Additional wording has been added in the text to clarify this.

14. page 22, line 4: "The target data was" should be "The target data were"

Corrected as stated.

15. Authors should clarify the time periods used for Tables 5 and 6. In particular, I'm a bit confused how the consecutive time periods come in - is there something different in saying targets are based on 15 consecutive 8-day periods versus targets were based on 144 consecutive days of requests, for example, or was there something distinct about the 8-day (or 120-day) periods that were somehow used?

As stated in the text 8 days was chosen as it is the preferred turn over period for fresh blood.

16. Table 5: Why is the first entry for S=0 under "Targets based on 8-day periods"?

This indeed was a typo and has been corrected.

17. It is not clear why authors feel that the multinomial simulations validate the numbers calculated using the Mallows' bound, especially for phenotyping, where Table 4 suggests 188,000 donors need to be phenotyped, while Table 6 has 95% probability at about half that size (94,156). Yes, the Mallows' bound value will exceed 95% probability based on the simulations, but that is potentially quite a bit more phenotyping than would be needed based on the simulations.

The text below has been added to further clarify the use of the given model. In addition, the analysis around tables 4, 5 and 6 was added as a cross validation for results based on the available data. The choice of the time periods is to match Lifeblood’s actual procedures, and this had already been noted in the text. The difference in the estimates is due to a confounding of the rare event probabilities the shorter time periods and the 95% success rate. The text that has been added in the discussion of the model on page 15 is copied below.

It is well known that one area in which standard Monte Carlo simulations do not work particularly well are in rare event simulation [10]. In this case many simulations may be needed to estimate these rare events, and this can become computationally extremely burdensome. New approaches have been developed such as the cross-entropy method [10] that can be represented as a stochastic search algorithm that can address the issue of rare event estimation, but implementations are not trivial. However, in our case we have access to tight upper and lower bounds on the multinomial probability of meeting targets with high probability through the theory of Mallows. Thus rare event estimation of rare antigen combinations is just a matter of computing marginal (binomial) probabilities and avoids the issues associated with rare event simulation through Monte Carlo approaches.

[10] R.V. Rubinstein, D.P. Kroese, Simulation and the Monte Carlo Method, John Wiley and Sons, 2008.

18. Authors should provide some discussion of strengths and limitations in their conclusions.

A discussion of strengths and limitations have been added to the conclusion.

---

## [Decision Letter · Decision Letter 3]

6 Sep 2022

PONE-D-21-20968R3Probabilistic mathematical modelling to predict the red cell phenotyped donor panel sizePLOS ONE

Dear Dr. Ginige,

Thank you for submitting your manuscript to PLOS ONE. After careful consideration, we feel that it has merit but does not fully meet PLOS ONE’s publication criteria as it currently stands. Therefore, we invite you to submit a revised version of the manuscript that addresses the points raised during the review process.

Please address the remaining issue(s).

We look forward to receiving your revised manuscript.

Kind regards,

Jeffrey Chalmers, Ph.D.

Academic Editor

PLOS ONE

Journal Requirements:

Reviewers' comments:

Reviewer's Responses to Questions

**Comments to the Author**

1. If the authors have adequately addressed your comments raised in a previous round of review and you feel that this manuscript is now acceptable for publication, you may indicate that here to bypass the “Comments to the Author” section, enter your conflict of interest statement in the “Confidential to Editor” section, and submit your "Accept" recommendation.

Reviewer #1: All comments have been addressed

Reviewer #2: (No Response)

2. Is the manuscript technically sound, and do the data support the conclusions?

Reviewer #1: (No Response)

Reviewer #2: Yes

3. Has the statistical analysis been performed appropriately and rigorously? 

Reviewer #1: Yes

Reviewer #2: Yes

4. Have the authors made all data underlying the findings in their manuscript fully available?

Reviewer #1: Yes

Reviewer #2: Yes

5. Is the manuscript presented in an intelligible fashion and written in standard English?

Reviewer #1: No

Reviewer #2: Yes

6. Review Comments to the Author

Reviewer #1: (No Response)

Reviewer #2: The authors have addressed the majority of my earlier concerns. The one remaining point has to do with the statement of Theorem 2, giving the bounds. I had raised this issue before and the author's response does not address my comment. The 2nd upper bound (given in line 17 on page 13) does not appear to be correct (looking at Mallows' paper, the exponent has a negative sign and the direction of the inequality also should be switched). The bound, as written, is not interesting. The exponent (sum of the probabilities), as written, has a minimum of 0, which means the smallest this upper bound can be is 1 (e^0=1). Since probabilities have to be less than or equal to 1, this bound, as written, does not really give us any information. I'm not even sure if this is the upper bound authors use in tables 3 and 4 or if they use the upper bound from line 16 (product of the probabilities). Authors need to either correct line 17 or remove it.

7. PLOS authors have the option to publish the peer review history of their article (what does this mean?). If published, this will include your full peer review and any attached files.

Reviewer #1: No

Reviewer #2: No

---

## [Author Response · Author response to Decision Letter 3]

26 Sep 2022

Thank you for the opportunity to resubmit the manuscript “Probabilistic mathematical modelling to predict the red cell phenotype donor panel size”. We would also like to thank the Reviewers for their careful reading of the article and their insightful comments. 

We unreservedly apologize to Referee #2 for our failure to make their important correction to our statement of Theorem 2.

Please see below for details where we have responded to the requirements of the journal and the reviewers’ comments; we list only the outstanding issues. We give responses to specific Reviewer comments in blue below.

Response to the Reviewers:

5. Is the manuscript presented in an intelligible fashion and written in standard English?

Reviewer #1: No

Reviewer #2: Yes

RESPONSE:

We believe, like Reviewer #2, that the manuscript “is presented in an intelligible fashion and written in standard English”. Moreover, Reviewer #1 has previously answered “Yes” to this question and has not presented specific criticisms for us to address.

6. Review Comments to the Author

Reviewer #1: (No Response)

Reviewer #2: The authors have addressed the majority of my earlier concerns. The one remaining point has to do with the statement of Theorem 2, giving the bounds. I had raised this issue before and the author's response does not address my comment. The 2nd upper bound (given in line 17 on page 13) does not appear to be correct (looking at Mallows' paper, the exponent has a negative sign and the direction of the inequality also should be switched). The bound, as written, is not interesting. The exponent (sum of the probabilities), as written, has a minimum of 0, which means the smallest this upper bound can be is 1 (e^0=1). Since probabilities have to be less than or equal to 1, this bound, as written, does not really give us any information. I'm not even sure if this is the upper bound authors use in tables 3 and 4 or if they use the upper bound from line 16 (product of the probabilities). Authors need to either correct line 17 or remove it.

RESPONSE:

We unreservedly apologize to Reviewer #2 for failing to correct this misprint

The inequalities in Theorem 2 now read

1-∑_(j=1)^r▒〖Prob(〗 〖n_j<k_j)≤P〗_S≤∏_(j=1)^r▒Prob(n_j≥k_j ) ≤exp(-∑▒Prob(n_j<k_j ) ).

Further Mallows states the inequality

1-∑_(j=1)^r▒〖Prob(〗 〖n_j<k_j)≤P〗_S≤∏_(j=1)^r▒Prob(n_j≥k_j ) 

which we use in our paper and references Esary JD, Proschan F, Walkup DW. Thus, we have included this paper in our References and changed the first citations of the Mallows paper to “Mallows (see also Esary, Proschan and Walkup)”.

In particular, we used ∏_(j=1)^r▒Prob(n_j≥k_j ) and the R language to compute the upper bounds presented in the paper. We checked our calculations by separately computing exp(-∑▒Prob(n_j<k_j ) ) using MATLAB although we didn’t report these values.________________________________________

---

## [Decision Letter · Decision Letter 4]

14 Oct 2022

Probabilistic mathematical modelling to predict the red cell phenotyped donor panel size

PONE-D-21-20968R4

Dear Dr. Ginige,

We’re pleased to inform you that your manuscript has been judged scientifically suitable for publication and will be formally accepted for publication once it meets all outstanding technical requirements.

Kind regards,

Jeffrey Chalmers, Ph.D.

Academic Editor

PLOS ONE

Additional Editor Comments (optional):

Reviewers' comments:

Reviewer's Responses to Questions

**Comments to the Author**

1. If the authors have adequately addressed your comments raised in a previous round of review and you feel that this manuscript is now acceptable for publication, you may indicate that here to bypass the “Comments to the Author” section, enter your conflict of interest statement in the “Confidential to Editor” section, and submit your "Accept" recommendation.

Reviewer #2: All comments have been addressed

2. Is the manuscript technically sound, and do the data support the conclusions?

Reviewer #2: (No Response)

3. Has the statistical analysis been performed appropriately and rigorously? 

Reviewer #2: (No Response)

4. Have the authors made all data underlying the findings in their manuscript fully available?

Reviewer #2: (No Response)

5. Is the manuscript presented in an intelligible fashion and written in standard English?

Reviewer #2: (No Response)

6. Review Comments to the Author

Reviewer #2: (No Response)

7. PLOS authors have the option to publish the peer review history of their article (what does this mean?). If published, this will include your full peer review and any attached files.

Reviewer #2: No

---

## [Editor Report · Acceptance letter]

18 Oct 2022

PONE-D-21-20968R4 

Probabilistic mathematical modelling to predict the red cell phenotyped donor panel size 

Dear Dr. Ginige:

I'm pleased to inform you that your manuscript has been deemed suitable for publication in PLOS ONE. Congratulations! Your manuscript is now with our production department. 

Kind regards, 

on behalf of

Dr. Jeffrey Chalmers 

Academic Editor

PLOS ONE